Telomere length and dynamics in Astyanax mexicanus cave and surface morphs

Lunghi Enrico 1
Bilandžija Helena hbilandz@irb.hr 2
1 Department of Life, Health and Environmental Sciences, University of L’Aquila , L’Aquila , Italy
2 Division of Molecular Biology, Ruder Bošković Institute , Zagreb , Croatia
Langille Barbara
Electronic publication date: 2024 Feb 28
Publication date: 2024
Volume: 12
Electronic Location ID: e16957
Received 2023 Sep 13; Accepted 2024 Jan 25
Copyright: ©2024 Lunghi and Bilandžija
Copyright year: 2024
Copyright holder: Lunghi and Bilandžija
License: This is an open access article distributed under the terms of the Creative Commons Attribution License, which permits unrestricted use, distribution, reproduction and adaptation in any medium and for any purpose provided that it is properly attributed. For attribution, the original author(s), title, publication source (PeerJ) and either DOI or URL of the article must be cited.
License URL: https://creativecommons.org/licenses/by/4.0/

Keywords: Adaptation, Cavefish, Cave animals, Senescence, Telomere, Aging biomarker, Somatic redundancy, Biomarker of environmental exposures

Funding: Tenure Track Pilot Project from the Croatian-Swiss Research Programme of the Croatian Science Foundation and Ecole Polytechnique Fédérale de Lausanne This study was financed by the Tenure Track Pilot Programme of the Croatian Science Foundation and the Ecole Polytechnique Fédérale de Lausanne and the Project TTP-2018-07-9675 EvoDark, with funds from the Croatian-Swiss Research Programme. The funders had no role in study design, data collection and analysis, decision to publish, or preparation of the manuscript.

==============================
Background

Telomeres are non-coding DNA repeats at the chromosome ends and their shortening is considered one of the major causes of aging. However, they also serve as a biomarker of environmental exposures and their length and attrition is affected by various stressors. In this study, we examined the average telomere length in Astyanax mexicanus, a species that has both surface-dwelling and cave-adapted populations. The cave morph descended from surface ancestors and adapted to a markedly different environment characterized by specific biotic and abiotic stressors, many of which are known to affect telomere length. Our objective was to explore whether telomere length differs between the two morphs and whether it serves as a biological marker of aging or correlates with the diverse environments the morphs are exposed to.

Methods

We compared telomere length and shortening between laboratory-reared Pachón cavefish and Rio Choy surface fish of A. mexicanus across different tissues and ages.

Results

Astyanax mexicanus surface fish exhibited longer average telomere length compared to cavefish. In addition, we did not observe telomere attrition in either cave or surface form as a result of aging in adults up to 9 years old, suggesting that efficient mechanisms prevent telomere-mediated senescence in laboratory stocks of this species, at least within this time frame. Our results suggest that telomere length in Astyanax may be considered a biomarker of environmental exposures. Cavefish may have evolved shorter and energetically less costly telomeres due to the absence of potential stressors known to affect surface species, such as predator pressure and ultra-violet radiation. This study provides the first insights into telomere dynamics in Astyanax morphs and suggests that shorter telomeres may have evolved as an adaptation to caves.

Introduction

Cavefishes represent one of the most successful groups of vertebrates that colonized the subterranean world, with more than 350 species from every continent (Niemiller et al., 2019). Subterranean habitats are semi-enclosed ecosystems characterized by specific environmental features, such as the absence of light, stable microclimate, and scarcity of food resources (Culver & Pipan, 2019). These features triggered the evolution of specific traits in species that settled there (Balart-García et al., 2023). Characteristic traits of subterranean species may be morphological (e.g., loss of eyes and/or pigmentation, elongation of appendages), physiological (e.g., lower metabolism, loss of circadian rhythm), or behavioral (e.g., increase in exploratory behavior, lower intraspecific agonistic behavior) (Christiansen, 2012; Lunghi et al., 2023; Mösslacher & Creuzé Des Châtelliers, 1996; Poulson, 1963). A recent study that compiled evidence from the available literature highlighted the higher longevity of subterranean species compared to their closely related aboveground species and hypothesized that this may be considered an additional adaptive trait for subterranean life (Lunghi & Bilandžija, 2022).

A widely used method to predict the lifespan of individuals is to measure telomere length (Bize et al., 2009). Telomeres are repeated sequences of non-coding DNA located at chromosome ends in all vertebrates and most metazoans (Gomes, Shay & Wright, 2010; Monaghan, 2010); their function is to protect chromosomes from degradation and fusion (Armanios & Blackburn, 2012). The shortening of telomeres is one of the main causes of cell senescence (Campisi & D’Addadi Fagagna, 2007; Haussmann, Winkler & Vleck, 2005; Takubo et al., 2010). Each time a cell divides, telomere replication is incomplete, resulting in the loss of part of the telomere sequence and the generation of two daughter cells with shorter telomeres (Campisi & D’Addadi Fagagna, 2007). When telomere length is reduced to a critical length, the cell ceases replication and enters senescence; this mechanism is the basis of aging in many organisms (Blasco, 2007; Hemann et al., 2001). However, having longer telomeres (or higher telomerase activity) does not necessarily facilitate longer lifespan (Gomes et al., 2011; Gorbunova et al., 2014). Environmental exposures and different stressors (e.g., oxidative stress) can increase telomere shortening and drive cells into senescence earlier than is expected for their biological age (Boonekamp et al., 2013; Chatelain, Drobniak & Szulkin, 2020; Epel et al., 2004). Telomere shortening is sometimes counteracted by telomerase activity, which can maintain and extend telomere length, helping to slow overall organismal aging (Anchelin et al., 2011; Blasco, 2007). Furthermore, limited telomere length is necessary to prevent development of cancer (Gomes et al., 2011) as programmed cell death is critical for removal of damaged and potentially tumorigenic cells (Elmore, 2007; Strasser & Vaux, 2020). Therefore, telomere length and organismal longevity is not in a perfect correlation, and the balance between longevity and cancer dictates an optimal interval for each specific animal group.

In this study, we measured average telomere length (hereafter TL) in Mexican tetra, Astyanax mexicanus, a model species in evolutionary biology (Jeffery, 2020). This species has two ecomorphs, one fully adapted to living in caves—cavefish (hereafter CF) and the other living in surface waters—surface fish (hereafter SF). The advantage of using A. mexicanus in evolutionary studies is that comparisons between conspecifics allow us to avoid potential biases due to the intrinsic life history or evolutionary history differences that characterize different species, regardless of how closely related they are (Ficetola et al., 2018; Michaux, Libois & Filippucci, 2005; Verberk, Siepel & Esselink, 2008). The two Astyanax ecomorphs are adapted to living in environments characterized by different typologies and magnitude of stressors; for example, surface fish are subjected to high predator pressure and exposed to UV-radiation, whereas cavefish live in nutrient-limited and possibly hypoxic environment (Culver & Pipan, 2019). Our hypothesis was that the two ecomorphs initially shared similar TL and that this particular trait underwent selective pressure during the process of specialization to different habitats (Culver & Pipan, 2019). For example, UV radiation and high predator pressure are expected to shorten telomeres whereas calorie restriction and hypoxia would have the opposite effect. Therefore, we expect to find divergence in TL between Astyanax CF and SF. If so, it can be an effect of difference in longevity between the two morphs, consistent with the observed higher longevity of cave animals compared to their surface relatives (Lunghi & Bilandžija, 2022). Alternatively, considering that TL shortening is significantly amplified by various stressors individuals are exposed to (Von Zglinicki, 2002), and that aboveground and belowground environments are characterized by different stressors, a difference in telomere length between SF and CF could be the effect of different selective pressure promoted by the two environments.

Materials & Methods

Fish samples

We performed three different experiments to identify possible divergence in TL between Astyanax CF and SF. The samples used for our experiments were unrelated, so we provide a separate dataset for each of them (Tables S1–S3). All fish used in our experiments came from the Jeffery lab; the stock of SF originated from the Rio Choy river, while the stock of CF from Pachón cave. Fish in our experiment were randomly chosen from laboratory stocks and were kept in running water in not-enriched 40 liters aquariums since the age of one. Eight to fifteen individuals per aquarium were cultured at 23–25 °C under a 14–10 h light–dark photoperiod and fed daily with tetra flakes occasionally supplemented by living invertebrates (Jeffery, 2020). This study was performed in accordance with University of Maryland Animal Care and Use Committee (IACUC #R-NOV- 18–59) Project 1241065-1.

Caudal fin experiment

We collected fin clips (a few millimeters from the ventral lobe of the caudal fin) from 43 fish (23 SF and 20 CF) of different age (20 1.5-year-old: fish, 20 6-year-old fish and 11 9-year-old fish) and tested the potential divergence in TL (both length and shortening) between the two ecomorphs. This first experiment aims to evaluate the use of fin tissue to study TL in Astyanax fish. This sampling method is simple, fast, and does not require sacrificing the fish, allowing for sustainable use of laboratory and wild animals. The dataset can be found in Table S1.

Fish organ experiment

In this experiment we wanted to assess the potential variability of telomere length and attrition between different fish organs from both surface and subterranean environments. We also aimed to assess the potential correlation between the TL of caudal fin and different organs, in order to evaluate the use of caudal fin TL as a proxy for the TL of the entire organism. Twenty-two fish (12 SF and 10 CF) of different age (two fish of 2.5, one of 3, three of 4, four of 5.5, 12 of 6.5 and four of 7.5 years of age) were euthanized in a solution of Tricaine methanesulfonate (>2%) (cat #: A5040; Sigma Aldrich, St. Louis, MO, USA), and the following organs were macrodissected: caudal fin, bladder, skin, brain, gills, heart, liver, muscles, and gonads. Our aim was to collect fish of various ages, no other criteria prior to euthanizing the fish was used. The dataset can be found in Table S2.

Gamete experiment

In this experiment we aim to investigate a possible correlation between the TL of the adult caudal fin and the gametes of the fish to determine if telomere attrition occurs after fertilization and during development. We obtained both fin clips and gametes (eggs or sperm) from 42 reproductive fish (16 SF and 26 CF) spanning various age groups. The distribution of ages included two fish at 2 years, five individuals at 3 years, 10 at 3.5 years, 13 at 4 years, five at 5 years, one at 6 years, and six at 6.5 years old. Spawning was induced by gradually changing the water temperature of the fish system (from 22 °C to 24 °C to 26 °C to 24 °C to 22 °C) for four consecutive days, and gametes were collected on the second and third nights. Fish were placed upside down in a carved sponge soaked in water. Release of gametes was induced by massaging the abdomen. Eggs were collected with a stainless steel spoon, and sperm were collected with microcapillary tubes (cat #: 9000105; Rankweil, Austria). A clip of the caudal fin was also collected before release. The dataset is provided as Table S3.

DNA extraction and real time qPCR

We measured telomere length using real time quantitative PCR as it is a widespread and the most cost-effective method (Lin et al., 2019; Lindrose et al., 2021). All the sampled tissues and gametes were stored in 500 µl of lysis buffer (Zymo Quick-DNA Miniprep). Samples were homogenized in Bead Ruptor 4 (Omni International Kennesaw, GA, USA) with ceramic beads. DNA was extracted following the Zymo Quick-DNA Miniprep protocol. DNA concentration was measured with fluorometer DS-11 FX (DeNovix, Wilmington, DE, USA) using the DeNovix dsDNA Broad Range Kit (cat #: KIT-DSDNA-BROAD-2). DNA concentration was adjusted to 15 ng/µl for all samples to improve qPCR readability and comparability of the results. This concentration was chosen after a series of qPCR runs with different concentrations (9, 0.9, 0.09, 0.009) aiming to: i) identify the lowest DNA concentration that guarantee high PRC efficiency (see below), while ii) having the peak of DNA amplification on average <15 Ct. This allowed us to use tissues from which low DNA concentration were obtained, and to avoid confounding amplification of TEL primer, as it naturally shows PCR amplifications >30 Ct. After testing DNA dilution on tissues from five different individuals, only the concentration of 9 ng/µl produced on average 15 Ct (see Table S4). Therefore, we arbitrarily chose to set the DNA concentration to 15 ng/µl (50% more) to ensure qPCR results for TEL primers below Ct 15. We run the real time PCR on BIO RAD CFX96. We used the following primers for telomere length (Vasilishina et al., 2019) TEL; forward (F) 5′-CGGTTTGTTTGGGTTTGGGTTTGGGTTTGGGTTTGGGTT-3′, reverse (R) 5-GGCTTGCCTTACCCTTACCCTTACCCTTACCCTTACCCT-3′ and designed our own for a gene used as reference (OCA2; forward (F) CAAGAACACTCTGGAGATGGAG, reverse (R) ACGCAGCTCGTCAAAGTT). Gene oca2 (ENSAMXG00000012753) was chosen as a reference following recommendations of Vasilishina et al. (2019) because it is a single-copy gene and allows us to estimate the relative average telomere length. Primers were designed in the second exon and the amplicon was 109 bp in length. We checked target specificity using primerBLAST tool (NCBI) and by melt curve analysis. We first performed the serial dilution test using at least three DNA concentrations and six of primers, to assess the primer concentration with the highest efficiency (i.e., between 95% and 105%; Sveca et al., 2015): for TEL we chose 10 µM (slope-3.4, r2 0.99, efficiency 96.36%) and for OCA2 1 µM (slope-3.21, r2 0.99, efficiency 104.73%). Each 10 µl reaction consisted of 5 µl of ITAQ Universal SYBR Green Supermix (cat #: 172-5122; Bio-Rad, Hercules, CA, USA) + 2 µl of RNA/DNA free water + 1 µl of primer F and 1 µl of primer R + 1 µl of DNA. Each sample was assayed in triplicates. We ran samples in 96-well plate and each plate had six negative controls (the DNA was replaced by RNA/DNA free water), three for each primer. We set the qPCR with the following cycles: 30 s at 95 °C, 40 cycles of 95 °C for 5 s, 60 °C for 30 s (data collection) followed by a dissociation stage. We checked the reliability between different runs using a randomly chosen 8 samples from 3 individuals and running three different qPCR plates placing the samples in different order; standard deviation of Ct was <0.15, therefore we considered results from different runs comparable.

Data analysis

We used the built-in software (Bio-Rad CFX Maestro 2.2; Hercules, CA, USA) to extrapolate the Ct values for TEL and OCA2 from each sample. qPCR plates were considered contaminated and consequently discarded if Ct of negative controls were <30 for TEL and ≠ 0 for OCA2. We assessed the presence of potential pipetting errors by comparing the Ct values of each triplets: those showing a difference in Ct ≥ 0.5 were discarded and the DNA from the same sample was reanalyzed in a new qPCR run. The average Ct of the three replicates for both TEL and OCA2 was used for subsequent calculations. We calculated the TEL/OCA2 ratio applying the following formula 2−(TELCt−OCA2Ct), to further standardize qPCR results allowing comparisons between different samples and machine runs (Vasilishina et al., 2019). The results of this formula were log-transformed (hereafter, logCt) to improve normal distribution.

We performed subsequent analyses and prepared the figures in R environment using packages lmer4, lmerTest and visreg (Breheny & Burchett, 2017; Douglas et al., 2015; Kuznetsova, Brockhoff & Christensen, 2016; R Development Core Team, 2021).

Experiment with caudal fins

We performed analysis of variance (ANOVA) in which we used the relative telomere length (logCt) as dependent variable, while the fish age and fish ecomorph (SF vs CF) were set as independent variables.

Experiment with fish organs

We built a generalized linear mixed models (GLMM) in which we used the logCt of the organs as the dependent variable, while the type of organ, ecomorph, and fish age were added as independent variables; we added the interaction between organ and ecomorph as further independent variable to assess the potential variability for each organ between the two ecomorphs. The identity of the individuals was used as a random factor because we obtained multiple organs from each individual. Likelihood ratio test was used to evaluate the significance of the variables in the GLMM. We evaluated the potential correlation of TL between the organs for each ecomorph.

Experiment with gametes

We built a GLMM in which we used the logCt of the two different tissues as the dependent variable, while sample type (caudal fin vs. gametes), sex and fish age served as independent variables. The identity of individuals was used as a random factor since two different samples of each individual were analyzed. The likelihood ratio test was used to evaluate the significance of the variables in the GLMM. Finally, we evaluated the potential correlation of TL between caudal fin and gametes for each ecomorph.

Results

Experiment with caudal fins

The average telomere length (TL) was significantly correlated with the fish ecomorph (df = 1, F = 10.479, P = 0.002), while no significant correlation with fish age (df = 1, F = 0.062, P = 0.774) was observed; CF had shorter TL compared to SF (Fig. 1A).

Figure 1 Telomere lengths in the two Astyanax mexicanus ecomorphs: Pachón cavefish (CF) and surface fish (SF) from Rio Choy river.

(A) Partial regression plots showing the log-transformed relative telomere length (TEL_OCA2 ratio) for the caudal fin of the two studied populations of A. mexicanus. Horizontal line represents mean values, while shaded box are 95% CI. (B) Partial regression plots showing the log-transformed relative telomere length (TEL_OCA2 ratio) for the studied organs in the two A. mexicanus ecomorphs.

Experiment with fish organs

We successfully extracted DNA from 165 organs: bladder (18), brain (22), caudal fin (22), gills (22), gonads (14), heart (21), liver (16), muscle (15), skin (15) (Table S1). Overall, TL was significantly correlated with ecomorph (F1,18.24 = 10.82, P = 0.004) and organ type (F8,129.20 = 4.96, P <0.001), while no significant effect was detected for age (F1,23.61 = 1.2, P = 0.284) and for the interaction between organ and ecomorph (F8,130.18 = 0.38, P = 0.93). The SF showed longer TL than CF, while among organs the bladder and skin had the shortest TL (Fig. 1B). We found six significant correlations of TL in different tissues for SF, while only one for CF (Table 1). In SF, the gill had the highest number of significant correlations with other tissues (4), followed by the caudal fin, brain, and heart (2 each), and muscle (1). In contrast, in CF, only the correlation between brain and liver was significant (Table 1).

Table 1 Correlations among the fish organs used in the fish organs experiment.

The correlation between organs in surface Rio Choy fish (SF) is shown above, and for Pachón cavefish (CF) below. Significant correlations are marked in bold.

Surface fish	Brain	Caudal fin	Gill	Gonads	Heart	Liver	Muscle	Skin	
Bladder	P = 0.317	P = 0.895	P = 0.029	P = 0.904	P = 0.508	P = 0.860	P = 0.78	P = 0.905	
	R =  − 0.38	R = 0.05	R =  − 0.72	R = 0.07	R =  − 0.28	R =  − 0.09	R = 0.15	R =  − 0.06	
Brain		P = 0.01	P = 0.049	P = 0.871	P = 0.02	P = 0.324	P = 0.89	P = 0.168	
		R = 0.71	R = 0.58	R =  − 0.08	R = 0.69	R =  − 0.40	R = 0.06	R = 0.58	
Caudal fin			P = 0.303	P = 0.23	P = 0.007	P = 0.62	P = 0.485	P = 0.232	
			R = 0.32	R =  − 0.52	R = 0.76	R = 0.21	R = 0.32	R = 0.52	
Gill				P = 0.314	P = 0.027	P = 0.758	P = 0.02	P = 0.334	
				R = 0.45	R = 0.66	R =  − 0.13	R =  − 0.83	R = 0.43	
Gonads					P = 0.281	P = 0.844	P = 0.775	P = 0.953	
					R =  − 0.53	R =  − 0.09	R =  − 0.15	R = 0.05	
Heart						P = 0.74	P = 0.346	P = 0.781	
						R =  − 0.15	R =  − 0.47	R =  − 0.15	
Liver							P = 0.259	P = 0.464	
							R = 0.49	R =  − 0.54	
Muscle								P = 0.089	
								R =  − 0.91	
Cavefish	Brain	Caudal fin	Gill	Gonads	Heart	Liver	Muscle	Skin	
Bladder	P = 0.594	P = 0.097	P = 0.891	P = 0.341	P = 0.793	P = 0.288	P = 0.632	P = 0.988	
	R = 0.21	R = 0.59	R =  − 0.05	R =  − 0.47	R = 0.1	R = 0.47	R = 0.22	R =  − 0.01	
Brain		P = 0.301	P = 0.124	P = 0.404	P = 0.84	P = 0.017	P = 0.224	P = 0.062	
		R = 0.36	R = 0.52	R = 0.38	R = 0.07	R = 0.8	R = 0.48	R = 0.68	
Caudal fin			P = 0.302	P = 0.719	P = 0.604	P = 0.26	P = 0.723	P = 0.987	
			R = 0.36	R = 0.17	R = 0.19	R = 0.45	R = 0.15	R =  − 0.01	
Gill				P = 0.329	P = 0.397	P = 0.613	P = 0.558	P = 0.22	
				R = 0.43	R =  − 0.30	R = 0.21	R = 0.24	R = 0.49	
Gonads					P = 0.675	P = 0.215	P = 0.257	P = 0.917	
					R = 0.19	R = 0.54	R = 0.50	R =  − 0.05	
Heart						P = 0.452	P = 0.89	P = 0.747	
						R = 0.31	R = 0.06	R = 0.14	
Liver							P = 0.114	P = 0.782	
							R = 0.6	R = 0.15	
Muscle								P = 0.766	
								R =  − 0.16	

Experiment with gametes

TL significantly correlated with ecomorph (F1,38 = 6.99, P = 0.012) and with sex (F1,38 = 6.18, P = 0.017), while no significant effect was observed for fish age (F1,38 = 0.1, P = 0.751) and tissue type (F1,41 = 3.84, P = 0.057), although significance is slightly above the limit set at 0.05 for the latter. The TL was longer in SF compared to CF (Fig. 2A), and in males compared to females (Fig. 2B). Caudal fin TL was significantly correlated with TL in gametes in SF (R = 0.6, P = 0.013), whereas no significant correlation was found between TL of these two tissues in CF (R = 0.36, P = 0.068).

Figure 2 Plots showing significant relationships in experiment with gametes.

(A) Partial regression plots showing the log-transformed relative telomere length (TEL_OCA2 ratio) for both gametes and caudal fin in the two A. mexicanus ecomorphs. (B) Partial regression plots showing the log-transformed relative telomere length (TEL_OCA2 ratio) in gametes and caudal fin for each sex.

Discussion

In this study, we observed a divergence in TL between the Pachón CF and Rio Choy SF (Figs. 1–2), where CF had shorter TL. This contradicts our prediction that CF could have longer TL than SF based on the expected higher longevity of cave animals (Lunghi & Bilandžija, 2022). However, we also observed no effect of age on telomere length in either of the two morphs. Considering this result and the lack of robust evidence for a divergence in lifespan between Astyanax SF and CF (Riddle et al., 2018; Simon et al., 2017), TL in A. mexicanus could be considered a proxy for somatic redundancy (i.e., the ability of an organisms to buffer exogenous stress) rather than a biomarker for aging (Boonekamp et al., 2013; Sauer et al., 2021).

CF having shorter TL than SF can be attributed to two possible explanations. Compared to subterranean animals, individuals from surface rivers are exposed to a wider variety of biotic and abiotic stressors, including oxidative stress due to faster metabolism, UV radiation, greater environmental fluctuations (e.g., microclimate), and predation risk (Culver & Pipan, 2019). All of these stressors can contribute to the acceleration of telomere shortening (Von Zglinicki, 2002). Therefore, longer telomeres in SF may be an efficient mechanism to counteract these stressors. On the other hand, subterranean organisms exhibit slower growth, reduced metabolism and limited reproduction rates (Bulog et al., 2000; Howarth & Moldovan, 2018; Poulson, 1963). These physiological adjustments correlate with slower telomere attrition. Furthermore, cave animals live in environments where they must cope with prolonged periods of starvation and hypoxia (Bizjak Mali, Sepčić & Bulog, 2013; Lipovšek et al., 2019; Van der Weele & Jeffery, 2022), conditions that may cause stress, but were shown to have a positive effect on telomere length (Iglesias et al., 2019; Wang et al., 2018). Therefore, the relief from some sources of stress present on the surface and the reduction of potential body damage in cavefish, may have contributed to relaxed selection and allowed CF to reduce TL. Alternatively, a direct selection for shorter TL may have occurred when surface ancestors colonized subterranean environments (Boonekamp et al., 2021; Gross, 2012). According to the “thrifty telomere” theory (Eisenberg, 2011), having longer telomeres is costly, and in nutrient-poor environments such as caves, where organisms have fewer resources to devote to their biological activities (Culver & Pipan, 2019) less costly and shorter telomeres could be beneficial and selected for. However, another possibility is that telomere length is controlled by pleiotropic effects of genes selected for a different phenotypic trait (Pathak et al., 2021) as is the case in the evolution of eye degeneration and albinism in Astyanax CF (Bilandžija et al., 2018; Bilandžija et al., 2013; Krishnan et al., 2022; O’Gorman et al., 2021; Yamamoto et al., 2009). Future research will enable assessing which of the evolutionary forces may be responsible for the divergence in TL between Astyanax morphs.

No significant effects of aging on telomere shortening in adult A. mexicanus suggest that telomere maintenance mechanisms may be very efficient in this species. Lost telomere sequences are mainly replaced by telomerase activity (Campisi & D’Addadi Fagagna, 2007; Gomes, Shay & Wright, 2010; Xie et al., 2008). Studies conducted on different fish species, reported that despite the high telomerase activity in different organs, there was a significant decrease in TL with the age, which is a clear indication of the senescence of the organism (Hartmann et al., 2009; Hatakeyama et al., 2008). In short-lived zebrafish, on the other hand, TL does not shorten until 24 months of age, implying that individuals avoid senescence for approximately four-fifths of their lives due to telomerase activity (Anchelin et al., 2011; Lau et al., 2008; Lund et al., 2009). In our model organism, A. mexicanus, we found no evidence of telomere attrition in adult individuals up to 9 years old, which is consistent with the estimated age of the oldest wild-caught A. mexicanus (Simon et al., 2017). Therefore, senescence in this species might not be correlated with TL shortening, but is regulated by other mechanisms (Gruber et al., 2014). However, we cannot rule out the possibility that A. mexicanus does not age significantly within the age range examined in our study (1.5–9 years) and that telomere attrition may occur in older fish like some other signs of senescence (Anchelin et al., 2011; Riddle et al., 2018). Also, our results were obtained from lab-reared fish and may require a careful interpretation (i.e., conditions in laboratory environments differ from those occurring at natural sites). However, this is the easiest and most precise way to obtain exact information on fish age. Furthermore, rearing in the laboratory conditions controls for the effects of differing ecology present in the two environments in nature and is ideal to uncover genetic differences between the two morphs.

The third experiment did confirm once again that SF have longer TL compared to CF (Fig. 2A), but also highlighted a significant difference between sexes (Fig. 2B), where males have longer TL compared to females. This is not surprising, as females usually invest more in the reproduction, which strongly impact the TL (Kokko & Jennions, 2008; Sudyka, 2019). Although the difference in TL between gametes and fin was slightly above the significance threshold (P = 0.05), a trend showing larger TL in the former can be observed (Fig. 2B). This suggests that telomere dynamics not only differs between the two morphs, but also between different ontogenetic stages.

We observed a clear divergence of TL between the fish organs investigated; in both ecomorphs, the shortest TL was observed in the bladder and skin. In bony fish, the bladder regulates buoyancy. It is composed mainly of collagen fibers and generally lacks vessels (Pough, Janis & Heiser, 2013). Therefore, oxidative stress and cell renewal in the bladder are likely to be very limited (Mizushima & Komatsu, 2011; Von Zglinicki, 2002), so reduction of mechanisms that counteract telomere attrition in this organ could reduce maintenance costs (Eisenberg, 2011). At the same time, the skin represents the protective layer that constantly defends organism against external stressors and therefore undergoes high cellular division, which accelerates the shortening of telomeres (Buckingham & Klingelhutz, 2011). On the other hand, other fish organs are involved in various biological processes such as metabolism, locomotion or information processing, and constant cell turnover and production of energy are essential for maintaining the vitality of these organs (Mizushima & Komatsu, 2011; Pough, Janis & Heiser, 2013), which means that the cells composing their tissues are constantly under oxidative stress, one of the major exogenous causes of telomere shortening (Eisenberg, 2011; Von Zglinicki, 2002). Therefore, prolonged TL in these organs may not only help maintain their vitality, but also increase overall survival and longevity of the fish (Boonekamp et al., 2013; Haussmann et al., 2002; Haussmann, Winkler & Vleck, 2005).

The TL of the different organs correlated poorly, showing only six significant correlations for SF and one for CF (Table 1). Of the total significant correlations, only two in SF involved the caudal fin. For these relatively small fish, fin clipping is a more practical and sustainable method of DNA sampling because it does not require sacrificing the fish and can be used in longitudinal studies due to the ability of Astyanax to regenerate fins (Stockdale et al., 2018). Our goal was to establish a non-lethal protocol for examining TL in Astyanax to identify potential variation in telomere attrition over the course of the lifespan of the fish. Unfortunately, TL of the caudal fin did not correlate significantly with other organs (especially in CF) in the populations we studied. However, repeating this experiment in other populations, especially sampled from natural habitats, may increase the possibility of developing a sustainable method for studying TL in Astyanax fish.

Conclusions

Our study provides the first information on the average telomere length in Astyanax mexicanus revealing that Pachón cavefish exhibit shorter telomeres compared to Rio Choy surface fish. Notably, we observed no evidence of telomere attrition as an indicator of senescence in adult fish used in our experiments, ranging in age from 1.5 to 9 years. While the laboratory conditions, marked by a stable environment, consistent food supply, and absence of predation risk (Chin et al., 2018; Chin et al., 2020), may have influenced these results, they also underscore the genetic basis of the observed telomere length differences between CF and SF. Under these favorable conditions, potential telomerase activity, known for mitigating telomere attrition due to environmental stress in the wild (Boonekamp et al., 2013) may have contributed to masking telomere attrition due to aging. Our analyses support the hypothesis that, at least in the laboratory stocks of A. mexicanus used in this study, average telomere length is more indicative of somatic redundancy than of individual lifespan. The less stressful (fewer predators, reduced environmental fluctuations) and longevity-promoting conditions (lack of UV irradiation, caloric restriction, hypoxia) in subterranean habitats may have contributed to cavefish exhibiting shorter telomeres that require less energy for maintenance. Additional analyses involving wild fish and other Astyanax populations will be important to validate and interpret these findings. Our study positions Astyanax mexicanus as a valuable model species for deepening our understanding of telomere dynamics and evolution.

Supplemental Information

Supplemental Information 1 ARRIVE 2.0 checklist

Table S1 Dataset used in Caudal fin experiment

ID = record ID; Ecomorph = surface (SF) or cave (CF) morph; Population = population code; Age = fish age; Average_Ct_TEL = average threshold cycle (qPCR) for telomere gene; Average_Ct_OCA2 = average threshold cycle (qPCR) for OCA2 gene; Delta_TEL_OCA2 = difference between the Ct of the two investigate genes; TEL_OCA2_ratio = TEL/OCA2 ratio obtained through the formula 2–(TELCt –OCA2Ct).

Supplemental Information 3 Dataset used in fish organs experiment

ID = record ID; Fish = fish identity; Ecomorph = surface (SF) or cave (CF) morph; Population = population code; Age = fish age; Organ = indicate to which fish organ corresponds the following data; Average_Ct_TEL = average threshold cycle (qPCR) for telomere gene; Average_Ct_OCA2 = average threshold cycle (qPCR) for OCA2 gene; Delta_TEL_OCA2 = difference between the Ct of the two investigate genes; TEL_OCA2_ratio = TEL/OCA2 ratio obtained through the formula 2–(TELCt –OCA2Ct)

Supplemental Information 4 Dataset used in Gamete experiment

ID = record ID; Fish = fish identity; Ecomorph = surface (SF) or cave (CF) morph; Population = population code; Sex = fish sex (male or female); Age = fish age; Tissue = indicate to which tissue corresponds the following data; Average_Ct_TEL = average threshold cycle (qPCR) for telomere gene; Average_Ct_OCA2 = average threshold cycle (qPCR) for OCA2 gene; Delta_TEL_OCA2 = difference between the Ct of the two investigate genes; TEL_OCA2_ratio = TEL/OCA2 ratio obtained through the formula 2–(TELCt –OCA2Ct).

Supplemental Information 5 The average of CT values from the serial dilution test aiming to assess the working concentration of DNA

DNA concentrations are expressed in ng/ μl, while each sample refers to a different fish.

Supplemental Information 6 MIQE checklist

Minimum Information for Publication of Quantitative Real-Time PCR Experiments checklist.

We thank WR Jeffery for kindly providing fish samples from his lab. We are very grateful to M Lukić and other members of our group for fish maintenance. We thank M Čupić for his help with gamete collection, and I Rubelj and L Nanić for discussions and suggestions.

Additional Information and Declarations

Competing Interests

Author Contributions

Animal Ethics

Data Availability

The authors declare there are no competing interests.

Enrico Lunghi performed the experiments, analyzed the data, prepared figures and/or tables, authored or reviewed drafts of the article, and approved the final draft.

Helena Bilandžija conceived and designed the experiments, authored or reviewed drafts of the article, and approved the final draft.

The following information was supplied relating to ethical approvals (i.e., approving body and any reference numbers):

University of Maryland Animal Care and

Use Committee (IACUC #R-NOV- 18–59)

The following information was supplied regarding data availability:

The raw data is available in the Supplementary Files.

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
