# Peer review of "Telomere length and dynamics in Astyanax mexicanus cave and surface morphs"

_PeerJ, doi:10.7717/peerj.16957_

## Round 0.1 · original submission · Major Revisions

Overall I really enjoyed the concept of the paper and the lab results are interesting and thought-provoking. However, there is a large issue in the formulation of the main hypothesis. Subterranean animals may live longer than surface conspecifics, that could be true, but I am not convinced based on Simon et al. that cavefish live longer. Based on table 1, only one cavefish site had longer living fish – all others matched surface fish ages. I think you need to go into more detail in your introduction on this uncertainty in the results or find different support for your theory. The particular cave that Simon et al. found to have the higher ages was not used in your study. Also, the lab-reared cave- and surface fish had the same ages. If you can find support for longer-lived cave animals in general (not limited to just cavefish), you should add it in, otherwise, your entire theory moving forward is not very strong. Also, I think the title needs to better reflect the actual results as you were unable to find evidence of longer lifespans in cavefish. Finally, the lab-reared fish are innately going to be different from wild fish. I realize the difficulty that comes with working with subterranean animals, but the lab environment is completely different than what surface and cavefish experience in the wild. Therefore, what do these results offer when thinking of how it can be applied back to wild fish - does it really provide a true assessment of telomere length and wild fish aging? I think you need to explore this more in your discussion.

·

Basic reporting

Throughout, the paper is written in a straightforward and concise style. Astyanax mexicanus is surely an interesting model to analyze adaptations to cave environments including telomere dynamics. Methods and procedures are sufficiently described, and the results are discussed thoroughly and in a broad context. Tables and figures are concise (but see one comment below) and have a satisfying lay-out. Raw data are provided as supplementary material.

Experimental design

The experinental design makes complete sense in my view.

Validity of the findings

I have some major and several minor issues:
Major issues:
1) In my view, the support for the premise that cave-populations are longer lived than surface-dwelling populations is currently too weak. The age differences cited from Simon et al. 2017 are negligible and thus, in my view, not sufficient to assume meaningful longevity differences between the two fish morphs. The other citation (Riddle et al. 2018) seems to provide somewhat better support on a qualitative level (signs of ageing), but this remains – as Lunghi and Bilandžija themselves admit – anecdotal; moreover, Riddle et al. 2018 state that “cavefish have a similar lifespan to surface fish”. The often cited longevity database AnAge has adopted this view. In sum, the evidence for a longevity difference between cave and surface populations is very vague and far from being convincing, which puts the whole rationale of the paper in question. I am not a fish reseracher but as far as I know, the model species is well-established and I assume that many labs word-wide have maintained these animals for a long time. I suggest that the authors contact fish holders and ask them about their experience concerning average and maximum lifespan of Astyanax mexicanus in captivity in order to support their premise more convincingly.
2) The second major issue I have is that the hypothesis (longer-lived morphs should have longer telmoreres and/or slower attrition), and more so its discussion, is in my view somewhat oversimplified. Telomere biology is complicated and so is any assumed connection between telomere length, attrition, telomerase activity and longevity. In mammals, replicative senescence due to telomere shortening is considered an anti-cancer-mechanisms and indeed high longevity has co-evolved with short telomeres (Gomes et al 2011, Aging Cell) and a reduction of telomerase activity in somatic cells (e.g. Gorbunova et al. 2014, Nat Rev Genet). Of course, fish are not mammals and the supectibility for certain cancer types probably differs between both taxa, but given that fish do develop neoplasias, potential paralelles between their findings and the mammalian patterns should at least be discussed by the authors.
3) The influence of age on telomere length is poorly represented. Most disturbing is that there are no graphs on the issue. Furthermore, information about age distribution in the sample is hard to find (nothing in the Method-section, vague information (“up to 9 years old”) is provided only relatively late in the discussion); the supplementary tables do not really help because the are badly formatted ( several cells contain calendaric dates instead of numbers); last but not least, the text contains contradictory statements, e.g. lines 229-231 (“no correlation between telomere length and fish ages”) vs. line 309 (“TL in SF incerased with age”).
4) It seems to me as if at least in some cases, references have been cited inappropriately. For example, Blasco 2007 and McLennan et al. 2018 are cited as references for the author´s statement that telomerase activity “can maintain and extend telomere length, helping to slow overall organismal aging.” However, Blasco 2007 is much more cautious in this regard (“demonstration of whether telomerase re-expression in adult tissues is able to […] extend the lifespan of organisms is still pending”), and McLennan et al 2018 did not even measure telomerase activty, nor did they write much about it. Also, LuIkiewicz et al. 2020 are much more cautious than the statement they were cited for in lines 53/54. I did not cross-check all references in the same detail but I suggest that the authors re-evaluate whether their statements and the references they chose to suppport them are congruent, and correct wherever they are not.

Additional comments

Minor issues:
1) Line 46: re-phrase to “loss of eyes and/or pigmentation”? The statement is on subterranean species in general, i.e. also includes e.g. subterranean mammals which reduced eyes, but did not loose pigmentation
2) Line 48: Lunghi et al. in press is not in the reference list
3) lines 266 – 268 seem to contradict the statement in lines 260 – 264 that exposure to stressors accelerates TL shortening
4) Line 271: statement is ambiguous; what is meant with “These activities”: metabolism, growth and reproduction, or their reduction?
5) Materials and methods: use of tempi (past and present tense) is not homogenous, e.g. lines 103-105 or 124/125
6) line 203: verb is missing
7) Lines 221/222: why is sex a random factor? There may be sex differences in TL length
8) line 324: variance values are not provided, so this statement is hard to evaluate

Reviewer 2 ·

Basic reporting

Positives: Overall, I think the introduction did a good job of explaining the reasoning for this study. I think the explanation of telomeres was good to help explain what they are and why they are being looked at. The expected results were clear.

Minor issues:
o 1: Do cave and surface populations have differences in lifespan in the lab? If this information is available, it would help to inform if longevity in naturae is primarily based on stressors affecting surface fish more than cave fish.
o 2: I think a figure that compares age and telomere length would be useful to visually see that information.
o 3: Raw data needs descriptors for what each variable represents.
o 4: Title needs to be reworked because it currently makes it sound like multiple species are being looked at and that multiple causes of longer lifespan are being investigated when only one species is being looked at and is focusing on just telomere length and shortening.
o 5: Sentence starting on line 30 with “This allows” needs to be restructured so that the meaning is clearer. I think a different word or phrase should be used other than “cheaper” to describe that shorter telomeres may be less energetically costly.
o 6: Line 26 might read better if instead of slower shortening it read “reduced rates of shortening”.
o 7: Sentence starting on line 103 with “Fish employed” needs to be rewritten because it is not very clear. Also, probably just use fish used instead of employed because that implies that they are paid.
o 8: Line 46 says that loss of eyes and pigmentation is adaptive but there is no consensus on if that is true
o 9: Line 63 the phrase “On the other hand” does not make sense in the context and is unneeded.
o 10: Possibly add information about what environmental stressors could be reducing telomere length in surface populations versus cave populations in the introduction. This information was talked about in the discussion but may be helpful in the introduction as well.
o 11: I am confused by the sentence starting on line 266 and what point that sentence is trying to make.
o 12: Increasing the size of the graphs or at least the labels would make reading the graphs easier.
o 13: On line 108 the term employment should be switched with a different word
o 14: Line 149 change beats to beads.
o 15: Line 155: need to add us after “This allowed”.
o 16: On line 290 need to add “us” after “allow”.
o 17: On line 356 change “harmless” to “non-lethal”.
o 18: Line 159: change arbitrary to arbitrarily.
o 19: On line 358 change the phrase “fish life” to “over the course of the lifespan of the fish” or something similar.
o 20: On line 283 change cheaper to a different word.

Experimental design

Positives: I think the comparison of caudal fin tissue to other tissue to analyze the effectiveness of that tissue was an interesting addition that helps to know the usefulness of that method.

Major issues:
o 1: Difficulty following the logic of the second hypothesis. The hypothesis states that aboveground populations will have more stressors and thus higher attrition rates, but the study is being done in the laboratory where the environments are the same. This makes sense if the study was done in the field, but in the lab the attrition rates should then be similar based on that reasoning.
o 2: Missing information in the methods about how different aged individuals were chosen and the number of individuals from different age classes.

Minor issues:
1: Would like to see more of an explanation on how qPCR can be used to measure telomere length. This would help readers not familiar with these methods follow along better.

Validity of the findings

Positives: I think this study is an interesting look into the aspect of aging and trying to determine why cave organisms live longer than those at the surface. I think this study can be used as a starting point to continue to use cave organisms as a model for aging.

Minor issues:
o 1: In the discussion, the result that TL increased with age for SF was presented for the first time but should have been stated at some point in the results.
o 2: Providing a brief explanation of somatic redundancy would help a broader audience understand the conclusions being made.
o 3: Might be worth mentioning that some of the cave environmental variables are stressful, but not in a way that reduces telomeres.

---

## Round 0.2 · accepted · Accept

The authors have done a fantastic job at addressing all the concerns and comments from the reviewers and myself. The paper is very clean and is a really interesting study. The only small changes that still need to be made are:

1) please use indents for different paragraphs in the introduction and methods (this could just be an issue with track changes but I can't see different paragraphs),

2) do you mean stainless steel spoon (missing the word steel from the sentence) on line 141,

3) the sentence on line 260 is awkward and could be rewritten, and

4) please remove the word 'necessarily' from line 335. These small changes are extremely minor and once those have been taken care of, the paper will be ready for publication.